# Functional Hydrogels: A Promising Platform for Biomedical and Environmental Applications

**DOI:** 10.3390/ijms26189066

**Published:** 2025-09-17

**Authors:** Mohzibudin Z. Quazi, Aaquib Saeed Quazi, Youngseo Song, Nokyoung Park

**Affiliations:** 1Department of Chemistry and The Natural Science Research Institute, Myongji University, Myongji-ro, Yongin 17058, Gyeonggi-do, Republic of Korea; mhzb1195@gmail.com (M.Z.Q.);; 2Department of Biochemistry, Government Medical College and Hospital, Jalgaon 425001, MH, India

**Keywords:** functional hydrogels, crosslinked polymer network, healthcare, biomedical applications, stimuli responsive, environmental applications

## Abstract

Functional hydrogels are a growing class of soft materials. Functional hydrogels are characterized by their three-dimensional (3D) polymeric network and high water-retention capacity. Functional hydrogels are deliberately engineered with specific chemical groups, stimuli-responsive motifs, or crosslinking strategies that impart targeted biomedical or environmental roles (e.g., drug delivery, pollutant removal). Their capacity to imitate the extracellular matrix, and their biocompatibility and customizable physicochemical properties make them highly suitable for biomedical and environmental applications. In contrast, non-functional hydrogels are defined as passive polymer networks that primarily serve as water-swollen matrices without such application-oriented modifications. Recent progress includes stimuli-responsive hydrogel designs. Stimuli such as pH, temperature, enzymes, light, etc., enable controlled drug delivery and targeted therapy. Moreover, hydrogels have shown great potential in tissue engineering and regenerative medicine. The flexibility and biofunctionality of hydrogels improve cell adhesion and tissue integration. Functional hydrogels are being explored for water purification by heavy metal ion removal and pollutant detection. The surface functionalities of hydrogels have shown selective binding and adsorption, along with porous structures that make them effective for environmental remediation. However, hydrogels have long been postulated as potential candidates to be used in clinical advancements. The first reported clinical trial was in the 1980s; however, their exploration in the last two decades has still struggled to achieve positive results. In this review, we discuss the rational hydrogel designs, synthesis techniques, application-specific performance, and the hydrogel-based materials being used in ongoing clinical trials (FDA–approved) and their mechanism of action. We also elaborate on the key challenges remaining, such as biocompatibility, mechanical stability, scalability, and future directions, to unlocking their multifunctionality and responsiveness.

## 1. Introduction

Nanoscale engineering has explored various opportunities to interact with biological systems at the molecular level [1,2]. The beginning of nanotechnology in the late 20th century marked a significant paradigm shift [3,4,5]. Especially in medicine, the development of nanoscale platforms has led to diagnosis and disease prevention with precision [6,7,8,9,10]. Nanotechnology has become a vital field with broad applications at the nanoscale [11,12,13,14,15]. This has steered research toward the design of nanocomposites and nanoscale drug carriers [16,17,18,19,20]. Furthermore, these designs have resulted in several nanotherapeutic systems, such as nanotubes [21], micelles [22,23], dendrimers [24,25,26], and hydroxyapatite nanoparticles [27], for advanced drug delivery [28]. The design of appropriate carriers for targeted therapy involves delivering genetic materials, nucleic acids, and encapsulated nanocomplexes to damaged cells and tissues [29,30,31]. Despite these significant advances, many challenges exist [32,33,34]. The inefficient performance of various nanocarriers is caused due to interference from biological substances. Their plasma instability and premature renal excretion hinder sustained drug release due to insufficient residence time [33,35,36]. Studies have shown that issues like poor biodistribution, lack of biodegradability, cytotoxicity, and unstable in vivo performance have slowed the successful translation of these systems into clinical trials [37,38,39,40].

An ideal nanotherapeutic system must show potential for not only effective delivery, but also minimizing side effects [41,42]. Subsequently, studies have suggested that an effective drug delivery platform should transport therapeutics through blood vessels without premature leakages [43,44,45]. The first generation of nanocarriers laid the groundwork for a new class of therapeutics. Moreover, liposomes, spherical vesicles made of phospholipid bilayers, were among the first to receive clinical approvals [39], due to their ability to solubilize hydrophobic drugs and modify pharmacokinetics. Likewise, polymeric micelles feature a stable core for drug encapsulation through their self-assembly of amphiphilic block copolymers. Primarily, these platforms rely on passive targeting, which is known as Enhanced Permeability and Retention (EPR) [46]. The EPR effect has illustrated that to reduce systemic toxicity and expose healthy tissues less, nanocarriers might preferentially accumulate in diseased areas selectively [47,48]. A wide range of nanocarriers has been developed to comply with this passive targeting method for unconventional therapeutics. However, significant challenges remain in translating them into clinical use. The EPR effect has shown high variability in different cases, which has resulted in passive targeting as an accordingly significant challenge. As a result, the percentage of the injected dose that accumulates in the target tissue is often low, which limits its therapeutic effectiveness [49].

As mentioned earlier, issues such as poor targeting and the biological barrier, alongside the inherent properties of many first-generation nanoparticles, pose substantial problems [50]. Materials such as quantum dots (QDs), carbon nanotubes, and inorganic metal nanoparticles often display significant cytotoxicity and non-biodegradability [51,52,53]. These characteristics may lead to potential bioaccumulation and chronic toxicity [54,55,56]. These combined challenges highlight that a new class of materials is essential. One that is effective at complying with the conditions, capable of smart interactions with the biological environment, and inherently biocompatible. To overcome these barriers and complexities, researchers have developed an interconnected 3D polymer network that mimics tissue-like textures, called a hydrogel [31,57,58,59,60]. Hydrogels are engineered with a polymer network by using natural or synthetic materials with a high degree of flexibility, owing to their large water content. Hydrogels have surfaced as an excellent platform as a result of their rational structure and functional design due to their ease in tuning gel chemistry (Figure 1). Based on earlier reported studies, we have delivered a general discussion of rational design strategies for functional hydrogels [61,62,63,64,65]. Having outlined the limitations of conventional nanodisperse systems, we next turn to the engineering strategies and component-level methodologies that define nanogels, thereby illustrating how these advanced hydrogels are designed to overcome the aforementioned challenges.

## 2. Rational Design Strategies for Functional Hydrogels

Hydrogels have a pliable consistency, tissue-like texture, high absorption, and significant water retention while maintaining their structural integrity [65,66]. Hydrogels have played a unique role in enabling the engineering of polymeric crosslinked nanomaterials with diverse structures [67]. Hydrogels have gained considerable attention as an innovative biomaterial with great potential across many promising applications. The biocompatibility of the monomer building blocks contributes greatly to avoiding serious inflammatory responses. Recently, several comprehensive review articles have been published discussing the synthesis, characterization, and applications of biodegradable hydrogels as carriers for drugs and nucleic acids [64,68,69,70,71,72,73]. The porous designs of hydrogels are ideal for encapsulating and releasing therapeutic molecules. These systems have been scaled down to the nanoscale, creating a new and powerful platform called nanogels or nanohydrogels [60,74,75,76]. Nanohydrogels have been widely designed using natural, synthetic, and hybrid polymers. Nanohydrogels possess a high surface-to-volume ratio that leverages more efficient cellular uptake. Hydrogels exhibit superior biocompatibility compared with liposomes. The potential use of hydrogels in regenerative medicine and controlled drug delivery surpasses that of standard nanocarriers, like liposomes, micelles, etc. [77].

Nanohydrogels are engineered from a wide range of natural polymers (chitosan, alginate, and hyaluronic acid) or synthetic polymers (poly(N-isopropylacrylamide), polyethylene glycol, etc.), that allow for precise tuning of their chemical and physical properties [78,79,80,81]. The flexibility of these hydrogels enables them to respond to stimuli by incorporating specific functional groups into their polymeric network [82,83,84]. Nanohydrogels have been engineered to undergo volume phase transitions, collapsing in response to specific triggers within a disease microenvironment. These triggers can be internal/external, such as acidic pH of a tumor, overexpressed enzymes, or changes in the redox potential. Regarding the external stimuli, the triggers can be exposure to heat, UV/NIR light, or magnetic fields [41,85,86,87,88,89,90]. Rational design strategies in hydrogel engineering generally rely on tailoring a material’s structure and chemistry to match the intended application. For instance, monomer and polymer selection allows for precise control over biocompatibility and degradation rates. Crosslinking approaches, whether physical or chemical, are optimized to balance mechanical stability with dynamic responsiveness. Stimuli-responsive elements, such as pH or temperature-sensitive groups, are incorporated to enable controlled release and adaptive behavior. Additionally, biofunctionalization with peptides, ligands, or nanomaterials enhances specificity and broadens functionality. These strategies collectively highlight how rational design can systematically improve the performance of hydrogels across biomedical and environmental domains [91,92]. Natural polymers, such as hyaluronic acid, chitosan, alginate, and dextran, are highly attractive due to their inherent biocompatibility, low immunogenicity, and biodegradability. The intrinsic properties of such polymers anchor to the targeted sites successfully by providing a built-in targeting mechanism. As earlier studies have reported, in cancer cells the CD44 receptor is highly expressed, and hyaluronic acid can easily bind with the overexpressed CD44 receptors [93,94], whereas synthetic polymers, like poly(N-isopropylacrylamide) (PNIPAM), polyethylene glycol (PEG), poly(lactic-co-glycolic acid) (PLGA), and PHEMA, are synthesized with precise control [95,96,97].

Moreover, the hybrid design approach can provide versatile functionality due to multiple components being combined with a robust design. The fine tunability of synthetic polymers and the biological benefits of natural polymers can yield robust functional systems [98,99,100]. In hybrid designs, the crosslinking method is used to design individual polymer chains into stable three–dimensional networks by either chemical or physical means. Physical crosslinking-based designs can often be reversible and sensitive to environmental changes. These strategies involve non-covalent interactions, such as hydrogen bonding and hydrophobic interactions. Furthermore, by modulating the physicochemical properties and network–building strategies, self-healing and injectable hydrogels can be enabled [101]. Chemical crosslinking involves the formation of permanent covalent bonds, resulting in a more mechanically robust and stable network. The density of these crosslinks is a key variable that can be modulated to control a hydrogel’s mesh sizes [70]. These factors help with the swelling ratio, mechanical strength, and the diffusion rate of the encapsulated drug. Due to the control over the molecular weight functionality, it allows for the fine-tuning of mechanical properties and stimuli-responsive behavior. Also, by employing biodegradable crosslinkers, i.e., disulfide bonds or ester linkages, the degradation rate of a hydrogel can be precisely programmed (Figure 2) [70,96,102].

The hydrophilic and often neutral surface charges (zwitterionic or charge-balanced surface modifications that result in an overall neutral surface potential) help to repel protein adsorptions; this approach could be a substantial step toward active control over drug delivery [104,105]. Such a phenomenon could result in reducing opsonization and tolerating rapid clearance by the mononuclear phagocyte system, often known as the “stealth” effect [106,107]. Perhaps the most sophisticated design element is the incorporation of stimuli-responsive moieties that transform a hydrogel into a smart vehicle, by integrating functional groups that can detect and respond to specific sites. Hydrophobic drugs, proteins, peptides, and nucleic acids (siRNA, mRNA, and pDNA) are susceptible to enzymatic attacks. To protect them from enzymatic degradation until they reach their selective destination, encapsulation of therapeutic agents within a hydrogel could be an exceptional strategy [108,109]. In terms of target pH–responsive systems, polymers containing acidic or basic groups have been used, as such moieties ionize in response to pH changes [88,89,110,111]. The acidic microenvironment in tumors causes electrostatic repulsion, network swelling, and drug release. In thermos-responsive gels, polymers with a lower critical solution temperature (LCST) at physiological temperature have been employed. The higher the LCST of the polymer, the greater the hydrophobicity that causes collapse while releasing its payload. This process can be triggered by localized hyperthermia [112,113]. Referring to the details discussed above, strategies in hydrogel design have led to innovative approaches towards biomedical applications. In the next section, we highlight functional hydrogels’ recent applications in healthcare.

## 3. Biomedical Applications of Functional Hydrogels

Functional hydrogels are used in numerous biomedical applications, such as tissue generation, lubricants, wound dressing, contact lenses, and advanced drug delivery [114]. In the initial studies from the report of Wichterle and Lim, hydrogel was first used as a tissue scaffold and contact lens [115]. Their biomedical applications were later discovered in the early 21st century as aqueous–swellable crosslinked polymeric networks [95,114]. Functional hydrogels are notable for their biomedical applications, due to their physiological stability, molecular recognition, easy functionalization, and low immunogenicity. Moreover, their chemical and physical properties are dependent on the polymer used. These characteristics allow for hydrogels to work as stimuli–responsive hydrogels. Polymers are either synthetic or natural in origin. In the case of a synthetic polymer, the chain length and degree of crosslinking can be adjusted. On the contrary, natural polymers are derived from polysaccharides or proteins [116]. Consequently, those polymers offer distinct functions while offering advantages, such as simple manufacturing and higher safety, which may lead to finding a way for their clinical usage. The numbers of molecules, peptides, antibodies, heparin, and heparan sulfate have been used to engineer hydrogels with molecular recognition functionality [117]. The role of functional hydrogels in biomedical applications is vast. The functional hydrogels’ building strategies, monomer building blocks, and types of polymers determine their multiple applications. In some cases, a synthetic polymer-based functional hydrogel lacks molecular recognition, whereas it is easier for a natural polymer-based hydrogel to recognize target sites [118,119].

Recently, several review reports have been published detailing the biomedical applications of functional hydrogels based on their material properties, additional functionalization, and intrinsic properties [120,121,122,123]. The emerging trend in these studies is the natural material-based hydrogel, due to their properties such as biodegradability, negligible adverse effects, and renal clearances. Hyaluronic acid (HA) is one of the natural polymers, widely used in the engineering of natural polymer-based hydrogels. The biodegradability of HA is very high, which is a significant concern within researcher communities due to the early degradation of HA; it may result as a barrier to its long-term application, along with its off–target delivery. To overcome this, Yeom et al. [124] reported the use of divinyl sulfone (DVS) and 1,4-butanediol diglycidyl ether (BDDE) as HA crosslinking agents to enhance structural stability and excellent tissue augmentation. The authors reported a significant stability improvement compared with earlier reports (Figure 3). Here, the authors used HA-HMDA (hexamethylenediamine) instead of using reactive crosslinkers, such as DVS and BDDE. The hydrogels were prepared by direct amide bond formation between HMDA and the carboxyl groups of HA, activated with EDC (1-ethyl-3-[3-(dimethylamino)propyl]carbodiimide) and HOBt (d 1-hydroxybenzotriazole monohydrate). Moreover, Ding et al. [125] reported another strategy to increase the tensile strength and toughness of HA hydrogels by introducing a catechol group. The experimental studies suggested that the injectable HA self-crosslinking hydrogel had excellent mechanical and elastic properties, which resulted in an HA filler with no inflammatory response, as reported earlier. In another report, Xu et al. [126] reported on polyacrylamide hydrogel membranes grafted through N, N′-methylenebisacrylamide over cellulose nanofibers by a free radical mechanism. The interconnection of the amide group of poly (acrylamide) and hydroxyl and carboxyl groups is present on the surface of cellulose nanofibers (Figure 4).

Functional hydrogels are an emerging material that has shown convincing success in clinical trials. We present in Table 1 a detailed discussion of the natural and synthetic component-based hydrogels that have been studied in clinical trials.

## 4. Environmental and Soft Robotics Applications of Functional Hydrogels

Heavy metals, such as Iron (Fe), Zinc (Zn), Copper (Cu), and Cobalt (Co), are essential in small amounts for maintaining human metabolism [127]. Previous studies have reported that heavy metal ions can easily form specific complexes with biomolecules. However, the non-biodegradable nature of heavy metals can lead to high toxicity [128,129]. Several heavy metal ions are known to form specific biomolecule complexes due to the presence of sulfur, oxygen, and nitrogen via chelation or coordination bonds. Metal ions, such as Hg^2+^, Pb^2+^, and As^3+^, may result in toxicological effects on the nervous system. Likewise, Cd^2+^, Cu^2+^, Hg^2+^, and Pb^2+^ may have the same effects on the kidneys or liver. These specific interactions between proteins, peptides, ligands, and metal ions were further studied by Aaron et al. [130] This group reported an efficient chemical synthesis method to produce a hybrid protein–polymer hydrogel. This hybrid hydrogel showcased precisely controllable, structural, and mechanical properties through the folding state of the proteins. The authors used N– and C–termini of pea metallothionein (PMTs) proteins as the crosslinkers. Similarly, heavy metals can build up in ecological systems, causing irreversible pollution. Researchers have developed several methods for the early detection and removal of harmful heavy metals, including ion exchange treatments, electrochemical technologies, and membrane filtration [131]. Similarly, Liao et al. [132] reported a highly porous phosphate-functionalized graphene hydrogel electrode material by a two-step process for uranium U^6+^ electrosorption. By using the H_2_O_2_ etching reaction, pores were created on graphene sheets. Further on, phytic acid was used as a gelator to assemble 3D interconnected macrostructures of graphene. The authors reported that a maximum electrosorption capacity of 545.7 mg g^−1^ at 1.2 V and pH 5 was obtained. In another study carried out by Dave et al. [133], a regenerable polyacrylamide hydrogel-based sensor functionalized with a thymine-rich DNA for the simultaneous ultrasensitive detection and removal of Hg^2+^ from water was reported (Figure 5). The unique properties reported for these hydrogel-based sensors are their resistance to nucleases and that they can be rehydrated from dried gels for storage and DNA protection. In this report, the authors detected that Hg^2+^ selectively bound between the two thymine bases, inducing a hairpin structure where, upon addition of SYBR Green I dye, green fluorescence was observed. Moreover, the authors reported that Hg^2+^ detection was enabled by the discovery of Hg^2+^-mediated T-T DNA base pairing, where the stability of the T-Hg^2+^-T base pair was higher than that of the thymine–adenine bond (Figure 5C). It is known that acrylamide selectively binds with Hg^2+^ via the amide nitrogen (Figure 5D). Recently, by free radical solution polymerization, two pullulan-based grafted hydrogels were synthesized to detect heavy metal ions in aqueous solutions. Pullulan-graft-poly(acrylic acid) and pullulan-graft-poly(acrylic acid-co-acrylamide) hydrogels were synthesized by Sonmez et al. [134] The group reported a 461% swelling capacity of pullulan-graft-poly(acrylic acid-co-acrylamide) with 169.79 mg g^−1^ of cadmium(II) ions adsorption. The unique property reported by the author is the reusability; it can be used a minimum of three times to remove cadmium (II). However, these strategies are limited by their lower efficiency, expensive equipment, time-consuming processes, and high labor costs. The unique physicochemical properties and customizable features of hydrogels have inspired their use in innovative fields like soft robotics [135,136,137]. Hydrogel-based designs in soft robotics have been extensively studied because of their easy-to-mold strategy. Soft robots require sensors that are soft, stretchable, and safe, along with preservable adaptivity. Researchers have reported on several sensors using stretchable conductors. But the flexibility, transparency, and biocompatibility that hydrogels can provide are unique compared to other reported biomaterials. Referring to this, Sun et al. [138] designed an ionic skin by using ionic conductors together with stretchable and transparent dielectrics to make soft actuators. The sensors deformed in response to the applied forces, producing signals that could be measured using voltages below 1 V. The pressure applied by the bending finger movement was readily detected by sensory sheets up to 1 kPa (Figure 6A,B). Similarly, Cheng et al. [139] designed a novel, low-cost hydrogel large-strain sensor that can be used with prefabricated soft robots. The process to attach the sensor requires only the application of thin silane-containing layers to the hydrogel and to the elastomer. Then, the sensor is simply placed on the elastomer and cured. This simple stick-on method can integrate soft hydrogels into soft robots. This straightforward stick-on method of attaching prefabricated sensors is comparable to attaching silicon strain gauges and metal foil to traditional rigid robot architectures.

In environmental applications, hydrogels are being studied for their ability to sense and remove pollutants from water. Similarly, recent developments in environmental remediation have seen hydrogels effectively applied to a variety of pollutant classes beyond heavy metals, particularly organic dyes, antibiotics, and microplastics.

Jung et al. [140] fabricated a pH-sensitive nanofibrillar hydrogel composed of chitin and cellulose crosslinked with citric acid. The authors reported that this pH-sensitive biopolymeric nanofibrillar hydrogel is an effective dye removal adsorbent. Citric acid was used as a green crosslinker for the fabrication of the hydrogel and for its structural stability, whereas to remove the anionic dye Acridine Orange and cationic Methylene Blue with capacities of ~372 mg/g and ~140 mg/g, the electrostatic attraction between the hydrogel and dye molecules was used as the main dye removal mechanism. In this study, the hydrogels showed over a 95% adsorption capacity for their removal performance after five reuse cycles. In another study, Ma and their group showcased a pillararene-based supramolecular polymer hydrogel for the removal of organic dyes from water. The pillararene poly(sodium 4-styrenesulfonate)-based hydrogel exhibited excellent adsorption properties for Eriochrome black T (EBT), with a much higher adsorption rate than that of activated carbon. Ma et al. [141] reported its excellent selectivity and recyclability, with an adsorption capacity of (~1818 mg/g) for the dye Eriochrome black T. While dye removal remains a prominent area of study, recent efforts have also expanded toward addressing pharmaceutical pollutants, particularly antibiotics, which pose persistent risks to ecosystems and human health. In a recent study, a widely used antibiotic known as oxytetracycline, which is commonly used in cattle for the treatment of bovine respiratory disease (BRD), was found in cattle farm wastewater. Taktak and their group synthesized a poly(2-dimethylaminomethyl methacrylate) hydrogel incorporating TiO_2_ nanoparticles and optimized it for the effective removal of oxytetracycline. The synthesized poly(DMAEMA)@TiO_2_ nanocomposite hydrogel achieved ~770.54 mg/g adsorption of oxytetracycline from real farmland wastewater. The resulting hydrogel supports six adsorption–desorption cycles and improves water quality while maintaining an over 80% efficiency. The hydrogel nanocomposite is an environmentally friendly material and an agent that can rapidly remove antibiotics [142]. Furthermore, regarding chemical contaminants, functional hydrogels are now being engineered to capture solid micropollutants, such as microplastics, which present unique challenges due to their durability and resistance to conventional treatment methods [143]. For addressing microplastic pollution, a bio-based composite hydrogel made from chitin nanofibrils and cationic lignin achieved adsorption capacities of up to ~1790.8 mg/g under a neutral pH, with excellent mechanical resilience and the near-complete removal of nanoplastics over multiple reuse cycles [144]. Additionally, a triple interpenetrating polymer network (IPN) hydrogel infused with Cu-POM nanoclusters that showed ~95–93% removal of PVC and PP microplastics at a near-neutral pH, along with photodegradation capabilities and good durability over multiple cycles, was proposed by Dutta et al. [145] The authors showed that a highly efficient copper substitute, polyoxometalate (Cu-POM) nanocluster, infused within a triple interpenetrating polymer network (IPN) hydrogel, comprising chitosan (CS), polyvinyl alcohol (PVA), and polyaniline (PANI) (referred to as pGel@IPN), could mitigate microplastic contamination in water. These studies illustrate that design strategies involving surface charge tuning, high porosity, and stimuli-responsive crosslinking permit hydrogels to address diverse pollutants effectively. Moreover, these hydrogels can be effectively used to treat a wide range of pollutants in real-world environments.

According to a comprehensive report by Zenab et al. [146] published in 2022, a limited number of hydrogels have emerged for the selective removal of heavy metal ions. Despite these customized functional hydrogels performing well at removing toxic metal ions, the selective removal of these ions from the environment in a controlled manner remains a significant challenge. Simultaneously, manufacturing these hydrogels at a large scale could be a challenge, as the reusability and sustainability of hydrogels are limited to the lab scale. The role of functional hydrogels in situ, in vitro, and in vivo have been highly studied, but their actual applications have been limited. To implement the use of hydrogels for healthcare and environmental applications, significant efforts have been invested in trying to break these barriers by taking studies from the lab scale to clinical trials. Referring to these studies, Figure 7 highlights the hydrogels, along with their types and components, that have been extensively used in clinical studies.

## 5. The Role of Functional Hydrogels in Clinical Settings

Hydrogels are known for their biocompatibility and tissue-like textures, which highlight their advancements in clinical studies [147]. Since hydrogel properties can be tuned through various chemical approaches, the clinical and translational progress of functional hydrogels is accelerating. As listed in Table 1, several ongoing and completed clinical trials of hydrogels show that they have been used in multiple clinical settings since the 1980s. Different hydrogel technologies have received regulatory approvals for cancer treatment and aesthetic procedures [148]. Beyond these uses, hydrogels are being studied in clinical settings for broader applications in drug delivery, cosmetics, tissue regeneration, bone grafting, and so on. In this section, we report on the FDA–approved clinical studies that are widely used in biomedical applications based on their mechanism of action and inherent properties.

These FDA–approved clinical examples demonstrate the significant advancements towards translating hydrogel-based materials into approved clinical therapies. Sustaining the clinical development progress while addressing the existing challenges will be crucial to exploring and unlocking the capabilities of functional hydrogels.

## 6. Challenges, Limitations, and Safety Considerations

Functional hydrogels have shown great potential in various biomedical and environmental applications. However, several notable challenges limit their wider use in clinical and industrial settings. A key issue is their stability and integrity in physiological and natural environments. Functional hydrogels are especially prone to degradation due to enzymatic reactions, caused by the nucleases present in biological fluids. This vulnerability can lead to their early disassembly, loss of function, and target specificity. Conversely, synthetic polymeric hydrogels are known for their high resistance, but may undergo incomplete degradation, leaving toxic byproducts and persistent environmental impacts. Another important limitation is the potential immunogenicity of hydrogel components, which can activate unintended pathways. Regarding DNA, naturally derived polymers like hyaluronic acid are generally considered biocompatible. However, chemically modified nucleotides, crosslinkers, or synthetic polymers may trigger inflammatory responses. In environmental applications, releasing DNA-based materials into ecosystems, especially on a large scale, raises concerns about horizontal gene transfer and unforeseen effects. Some case studies have indicated that using cationic polymers to bind and deliver nucleic acids can damage membranes and cause inflammation. Overall, the development of nanomedicine faces challenges, such as reproducible large-scale manufacturing and adherence to good manufacturing practice (GMP). Likewise, batch-to-batch consistency and long-term stability with scalability also exist as significant barriers. Moreover, hydrogels’ mechanical properties and novel mechanisms of action require extensive testing to meet strict FDA and EMA safety standards. Additionally, toxicology, pharmacokinetics, and long-term biodistribution studies are necessary to assess their risk profiles.

Moreover, functional hydrogels often face challenges, such as burst release, unpredictable biodegradation, and immunogenic responses, that compromise therapeutic outcomes. For instance, PEG-based injectable hydrogels have shown rapid clearance and local inflammation in animal studies, limiting their long-term stability in vivo. While natural hydrogels such as collagen or hyaluronic acid provide excellent bioactivity, their poor mechanical strength and fast degradation hinder their utility as scaffolds. Hybrid or composite systems with synthetic polymers are increasingly required to ensure durability and tunable degradation. In the context of environmental water purification, hydrogel-based adsorbents functionalized with carboxyl or amine groups have shown promising roles in the removal of heavy metals, such as Pb^2+^ and Hg^2+^. However, they are challenged by their limited selectivity in complex wastewater matrices and difficulties in regeneration. Likewise, the lack of a comprehensive understanding of hydrogel–host interactions at the molecular and systemic levels remains a major scientific hurdle. In hydrogel-based studies, the landscape mapping of the materials’ interaction with the extracellular matrix, immune system, and microbiota/environmental microbiomes have not been completely elucidated. Such gaps make it difficult to predict the off-target effects or long-term outcomes. Therefore, as functional hydrogels stand at the frontier of advanced therapeutics and environmental solutions, their strategic design, comprehensive preclinical validation, and detailed mapping are essential for addressing these challenges and ensuring their safe, effective, and sustainable applications.

## 7. Future Perspectives

The research on functional hydrogels continues to advance; one of the promising directions is the use of bio-orthogonal chemistry. The inclusion of bio-orthogonal chemistry to design and modify functional hydrogels in situ without interfering with the biological processes might show a benefit for tackling the earlier-mentioned challenges. Earlier reports have suggested that click chemistry reactions, such as strain-promoted azid–alkyne cycloaddition, enable the conjugation of ligands, peptides, and therapeutic molecules to hydrogel matrices under mild conditions [149,150]. Such strategies have facilitated modular hydrogel assembly that has helped researchers to tune the properties dynamically at the point of care. Another emerging trend is the incorporation of hybrid nanocomposite materials. A combination of hydrogels with nanostructures such as graphene oxide, carbon nanotubes, or MOF (metal–organic framework) could impart additional functionalities, such as electrical conductivity, photothermal responsiveness, or catalytic activities, that could result in broadening the scope of applications.

The integration of 3D and 4D printing technologies with functional hydrogels has also garnered substantial attention. These advanced bioprinting methods now enable the precise spatial patterning of multiple cell types and bioactive factors within hydrogel matrices. This strategy has enabled patient-specific tissue constructs with complex architectures [151]. In recent times, researchers have demonstrated 4D-printed hydrogels that can change their shape and mechanical properties in response to stimuli. This advancement suggests the transformative potential of minimally invasive medical devices and adaptive scaffold engineering. Regarding the future outlook and the broader scope of today’s highlighted terms, artificial intelligence (AI) holds tremendous promise. The convergence of hydrogels with synthetic biology and AI for the rational design of next-generation therapeutic materials could help to explore the multifunctionality of hydrogels. Recent advances underscore the practical implementation of emerging strategies in functional hydrogel design. AI-driven approaches have been employed to predict hydrogel degradation kinetics and optimize mechanical properties, thereby accelerating the discovery of application-specific hydrogel formulations. Whereas 3D bioprinting technologies have enabled the fabrication of patient-specific hydrogel scaffolds with controlled architectures for cartilage and bone regeneration, the advent of 4D-printed hydrogels, capable of shape transformation in response to stimuli, has opened new possibilities in dynamic tissue engineering and soft robotics. Alongside these, bio-orthogonal chemistry has been harnessed for precise, in situ crosslinking of injectable hydrogels, ensuring spatiotemporal control without interfering with the native biological processes. These representative advances illustrate how cutting-edge strategies are enabling a transition from theoretical concepts to tangible platforms. These strategies are also reinforcing the potential of functional hydrogels for both biomedical and environmental applications [152]. Synthetic biology has the potential to integrate living cells, designed to detect and release therapeutic substances, directly into hydrogel structures, resulting in the dynamically termed “living materials” [153]. This strategy could be explored as an exceptional capability for responding to environmental cues with high specificity. From a broader perspective, the future of functional hydrogels will likely emphasize sustainability, regulatory harmonization, and accessibility. By predicting the possible challenges of material properties, such as degradation profiles and biological interactions, AI-driven computational modeling may accelerate the optimization of hydrogel formulations more efficiently [154]. In conclusion, advanced functionalization strategies are rapidly transforming hydrogels from static carriers into smart programmable platforms with multifaceted capabilities. As interdisciplinary partnerships are favored extensively, such advanced approaches may significantly impact environmental sustainability. Moreover, as new technologies are incorporated into hydrogel development, these substances could help to impact biomedical treatments and beyond.

## Figures and Tables

**Figure 1 ijms-26-09066-f001:**
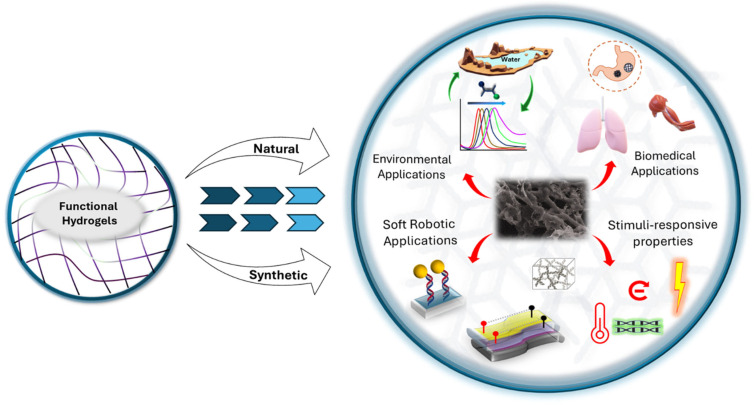
Graphical illustration of functional hydrogels and their advanced applications.

**Figure 2 ijms-26-09066-f002:**
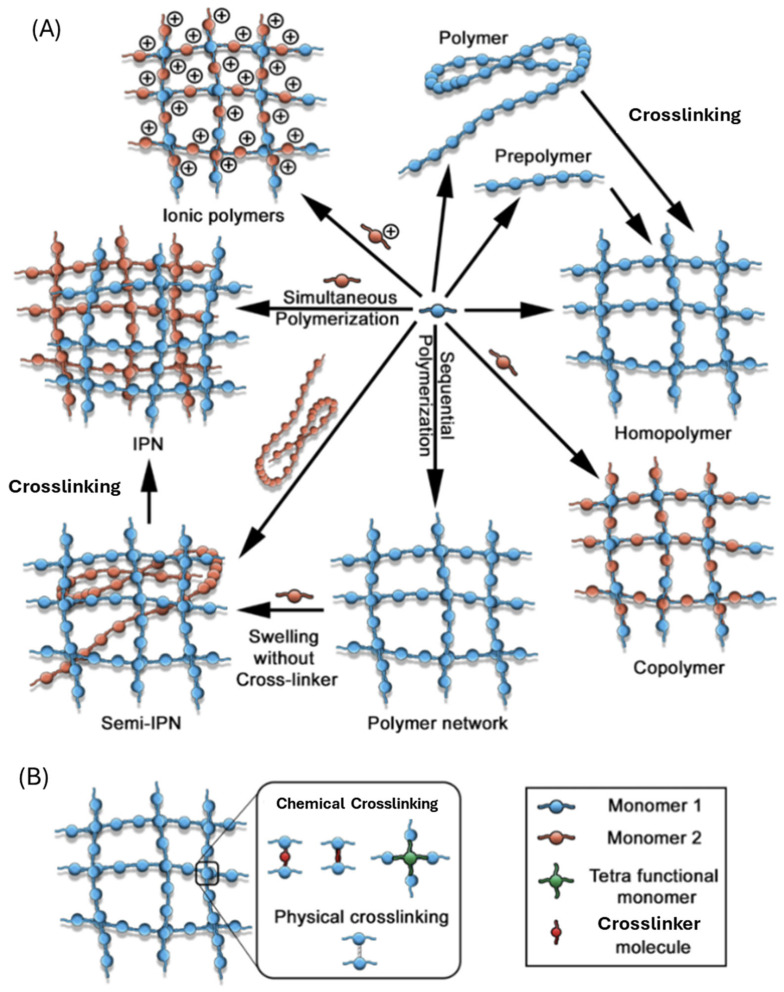
(**A**) Schematic representation of hydrogel engineering. (**B**) Types of crosslinking between polymer chains in a hydrogel. Reprinted with permission from reference [103] (copyright @Wiley 2020).

**Figure 3 ijms-26-09066-f003:**
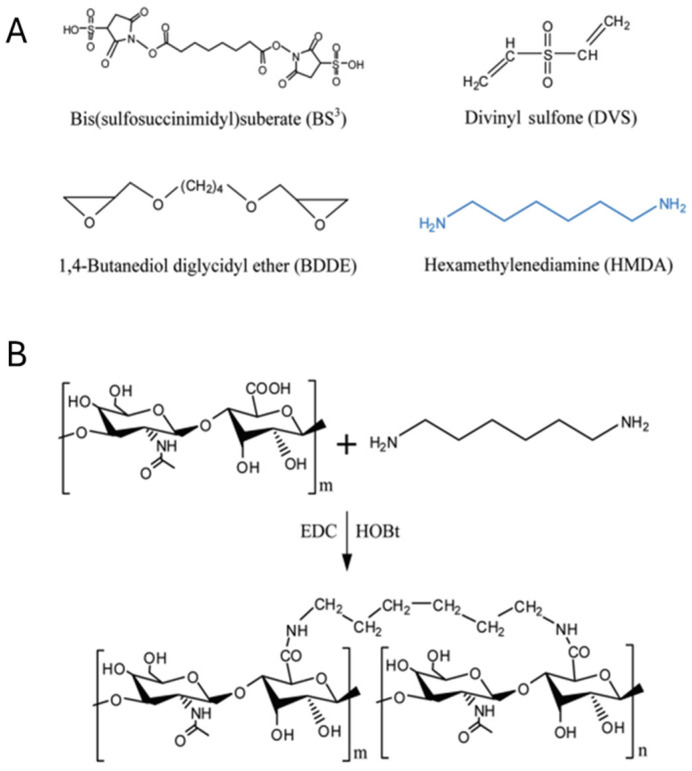
Schematic representations of (**A**) crosslinkers for the preparation of hyaluronic acid (HA) hydrogel dermal fillers and (**B**) for the preparation of HA hydrogels crosslinked with hexamethylenediamine (HMDA) after activation with 1-ethyl-3-[3-(dimethylamino)propyl] carbodiimide and 1-hydroxybenzotriazole monohydrate. Reprinted with permission from [124] (copyright (2010) American Chemical Society).

**Figure 4 ijms-26-09066-f004:**
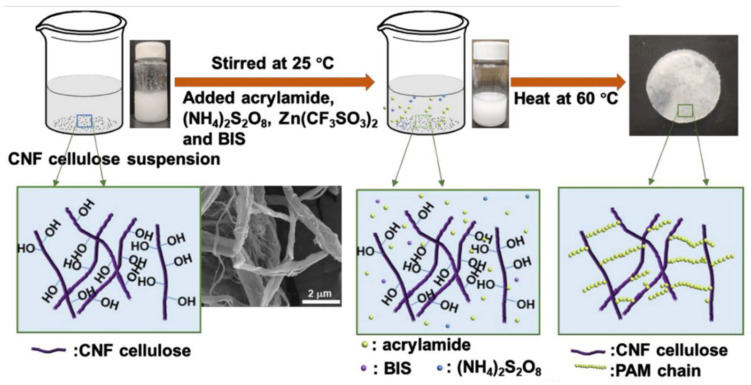
Schematic of synthesis route to form solid-state electrolytes by grafting PAM on cellulose nanofibers (CNFs) via a facile free radical polymerization approach. Reprinted with permission from [126] (copyright @Royal Society of Chemistry).

**Figure 5 ijms-26-09066-f005:**
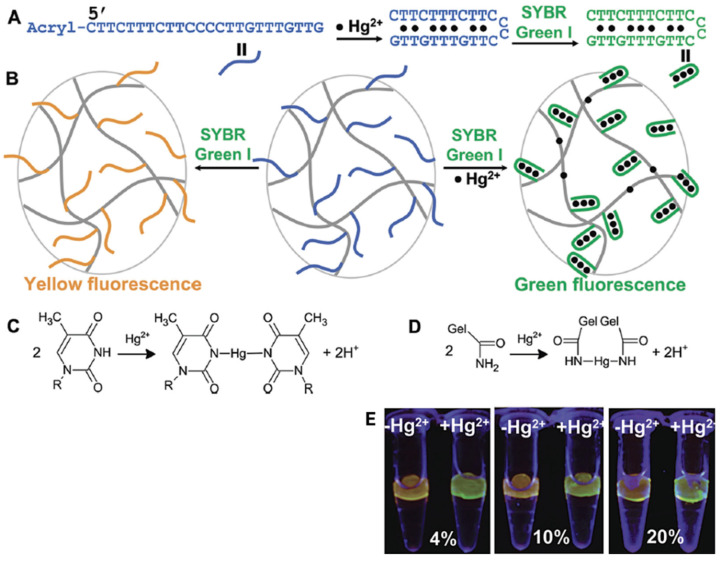
(**A**) DNA sequence of acrydite–Hg–DNA and fluorescence signal generation for Hg^2+^ detection. The 5′ end is modified with an acrydite group for hydrogel attachment. (**B**) Covalent DNA immobilization within a polyacrylamide hydrogel and interaction with Hg^2+^ and SYBR Green I produce a visual fluorescence signal. (**C**,**D**) Chemical reaction schemes of Hg^2+^ binding with thymine base pairs (**C**) and polyacrylamide in hydrogel (**D**). (**E**) Hg^2+^ detection as a function of gel percentage. Reprinted with permission from reference [133] (copyright (2010) American Chemical Society).

**Figure 6 ijms-26-09066-f006:**
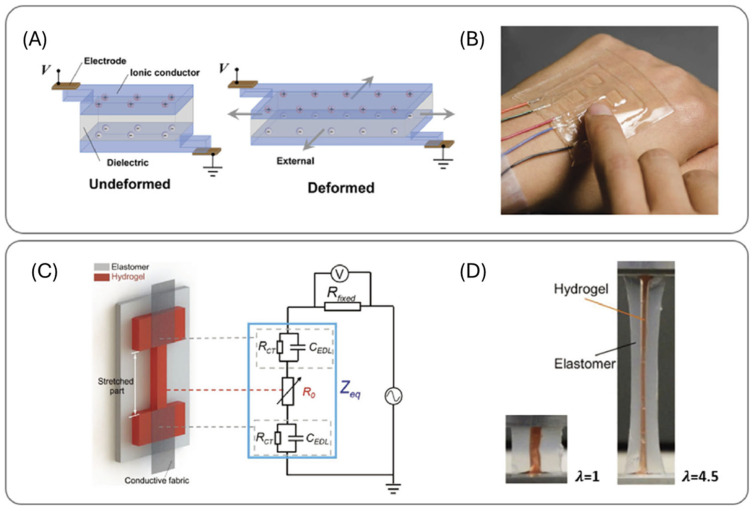
(**A**) Schematic diagram showing the working mechanism of capacitive stress/strain sensors. The capacitance between the two hydrogel electrodes increases when the sensor is deformed by external forces. (**B**) The transparent and stretchable sensor array operating on the back of a hand. (**C**) Schematic illustration of resistive strain sensors. The resistance of the hydrogel increases when the sensor is stretched by external forces. (**D**) The sensor can measure a strain of up to 450%. Reprinted with permission from references [138,139] (copyright @Wiley 2014 and 2019).

**Figure 7 ijms-26-09066-f007:**
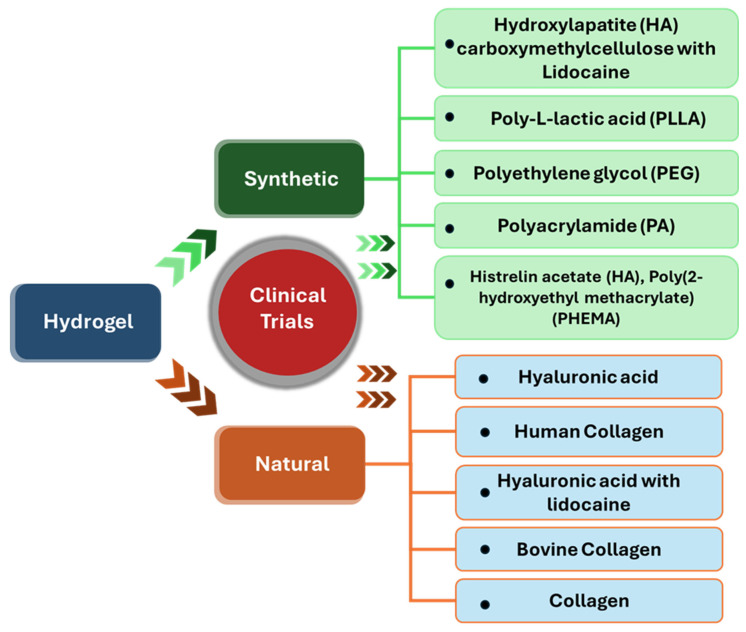
Types of hydrogels and hydrogel–forming biomaterials with components extensively used in clinical studies.

**Table 1 ijms-26-09066-t001:** Types of hydrogels and their clinical status with reported indications.

Sr No.	Hydrogel Type and Building Components	Mechanism of Action	Applications	Clinical Status
1	**Synthetic** *Histrelin acetate, poly* *(2-hydroxyethyl methacrylate), poly(2-hydroxypropyl methacrylate), and gonadotropin-releasing hormone*	The chemical reaction involves the interaction of histrelin with these polymers to form a hydrogel matrix, allowing for the controlled diffusion of the drug over time.	Palliative treatment of prostate cancer.	Approved—FDA 2004
2	**Synthetic** *Polyethylene glycol (PEG)*	(i) The hydrogel solidifies after its injection and creates a temporary spacer, creating a protective buffer zone to shield the rectum from the harmful effects of radiation while allowing the radiation to target the prostate cancer cells effectively. (ii) In cataract surgery it works as a sealant by creating a temporary, soft, and lubricious surface barrier to prevent fluid leakage from incisions following cataract surgery. (iii) The iodine from the gel provides a radiopaque property. The hydrogel functions primarily as a radiopaque tissue marker; visible hydrogel markers help guide radiation delivery, ensuring accurate targeting.	Prevention of tissue damage from prostate cancer radiotherapy. Sealant for wound healing and to prevent fluid leakage during surgeries. Assists with better soft tissue alignment for image-guided therapy.	Approved—FDA 2015 Approved—FDA 2014 Approved—FDA 2013
3	**Synthetic** *Hydroxylapatite carboxymethylcellulose with lidocaine*	The hydroxylapatite carboxymethylcellulose (CaHA) microspheres, once injected, act as a scaffold, encouraging fibroblasts (cells that produce collagen) to produce new collagen and elastin.	(i) Correction of wrinkles. (ii) Correction of wrinkles and folds; stimulation of natural collagen production.	Approved—FDA 2006 Approved—FDA 2015
4	**Synthetic** *Polymethylmethacrylate beads, collagen, and lidocaine*	Provides a long-lasting augmentation effect by stimulating the body’s collagen production.	Correction of nasolabial fold (smile lines).	Approved—FDA 2006
5	**Synthetic** *Histrelin acetate and poly(2-hydroxyethyl methacrylate) (PHEMA)*	The hydrogel allows for the controlled release of histrelin acetate (gonadotropin-releasing hormone). It helps to reduce sex hormones and delays the onset of puberty.	Central precocious puberty.	Approved—FDA 2007
6	**Synthetic** *Poly-L-lactic acid* (*PLLA)*	*PLLA* microparticles are recognized by the body as a foreign substance, triggering a controlled inflammatory response. This response activates fibroblasts, which are the cells responsible for producing collagen and elastin.	Restores facial volume and fat loss sign correction.	Approved—FDA 2004
7	**Synthetic** *Polyacrylamide (PA)*	*PAHG* works by creating a bulking effect within the urethra to improve urethral coaptation and reduce urine leakage associated with stress urinary incontinence. (ii) PA integrates with the synovium (joint lining) and forms a hydrogel matrix that provides long-term mechanical support and reduces friction by creating a cushioning and lubricating effect within a joint.	Improvement in Female Stress Urinary Incontinence (SUI). Osteoarthritis.	Approved—FDA 2006 Approved—FDA 2014
8	**Synthetic** *Calcium hydroxylapatite (CaHA), sodium carboxymethylcellulose (CMC), and glycerine*	CaHA particles provide immediate soft tissue augmentation. CaHA creates a bulking effect and helps to close the urethra. The increased bulk effect helps to reduce or eliminate SUI.	Improvement in Female Stress Urinary Incontinence (SUI).	Approved—FDA 2005
9	**Synthetic** *Porcine enamel matrix* *derivative in propylene* *glycol alginate gel*	It attracts mesenchymal cells to the root surface, stimulating the formation of new cementum, periodontal ligament, and bone. Promotes angiogenesis.	Periodontal tissue regeneration. Periodontal ligament reattachment.	Approved—FDA 1996
10	**Synthetic** *Calcium phosphosilicate* *particles, a PEG, and* *glycerine gel-like binder*	Calcium phosphosilicate stimulates bone growth, while the PEG and glycerine help maintain the material’s structure and facilitate its delivery.	Dental bone regeneration.	Approved—FDA 2005
11	**Synthetic** *Phase-pure silicon-substituted calcium* *phosphate in poloxamer 407 (Si-CaP)*	A synthetic triblock copolymer, poloxamer 407, creates a composite material that combines the bone-forming potential of Si-CaP.	Bone regeneration; bone void filler in orthopedic conditions.	Approved—FDA 2018
12	**Synthetic** *Demineralized bone matrix in poloxamer*	A bone graft material where (demineralized bone matrix) DBM, a collagen matrix with growth factors derived from human bone, is combined with a poloxamer 407 carrier. These components help provide the osteoconductive and osteoinductive signals to promote bone regeneration.	Bone regeneration; bone void filler in orthopedic conditions.	Approved—FDA 2005
13	**Synthetic** *Allographic demineralized bone matrix in polyethylene oxide polypropylene oxide block copolymer*	A carrier material made of a polyethylene oxide (PEO) and polypropylene oxide (PPO) block copolymer combined with DBM; this combination creates osteoconductive signals for bone regeneration in spinal fusion.	Void filler and graft regeneration.	Approved—FDA 2011
14	**Synthetic** *Allographic demineralized bone matrix in glycerol*	The DBM is processed to expose natural growth factors like BMPs (Bone Morphogenetic Proteins). The DBM provides a scaffold for new bone formation and releases growth factors that stimulate bone-forming cells to differentiate and build new bones.	Promoting bone health; bone graft extender and void filler.	Approved—FDA 2005
15	**Synthetic** *PEG 8000, dicyclohexyl* *methane-4, 40-* *diisocyanate, and* *1,2,6-hexanetriol/* *dinoprostone*	Dinoprostone stimulates the uterine muscles, leading to contractions similar to those in labor. It also promotes cervical ripening by activating the enzyme collagenase, which breaks down the collagen fibers in the cervix, causing it to soften, efface, and dilate, preparing it for delivery.	Cervical ripening and labor induction.	Approved—FDA 1993
16	**Synthetic** *PEG 1000, polyvinyl alcohol, and rice starch/* *Ondansetron*	Blocks the action of serotonin (5-HT) at the 5-HT3 receptors located in the chemoreceptor trigger zone (CTZ) of the brain and peripherally on the vagal nerve terminals in the gastrointestinal tract.	Nausea and vomiting control caused due to chemotherapy and radiation.	Approved—FDA 1991
17	**Synthetic** *PEG ester, trilysine amine,* *and decahydrated sodium* *borate*	The PEG ester and trilysine amine are the precursors, and the sodium borate acts as an accelerator, facilitating the crosslinking process. The hydrogel utilizes the crosslinking of PEG ester and trilysine amine triggered by the decahydrated sodium borate.	To seal the dura mater, the membrane surrounding the brain and spinal cord, during surgeries.	Approved—FDA 2005
18	**Synthetic** *Silicone*	(i) The silicon-made hydrogel lenses flatten the central corneal tissue and create a mid-peripheral steepening. That helps to slow the axial length elongation of the eye. (ii) The incorporation of phosphorylcholine (PC), a biomimetic molecule, creates a uniform physiological surface that attracts and binds water molecules, mimicking the natural tear film. This resists protein and lipid deposition, reducing eye dehydration. (iii) A viscous, cohesive gel that mimics the texture of natural breast tissue and maintains its shape even if the shell is ruptured.	Hyperopia/ myopia management. Myopia. Breast implants.	Approved—FDA 2018 and 2021 Approved—FDA 2019 Approved—FDA 2013 and 2014
19	**Synthetic** *2-Hydroxyethylmethacrylate,* *2-methacryloxyethyl* *phosphorylcholine, and* *ethylene glycol* *dimethacrylate*	It works by mimicking the properties of natural tears, specifically by phosphorylcholine (PC). This helps to maintain hydration and comfort.	Visual acuity.	Approved—FDA 2013
20	**Synthetic** *Cellulose and citric acid*	The expanded hydrogel particles, mixed with food, increase the volume of the stomach contents, creating a sense of fullness.	Obesity.	Approved—FDA 2019
21	**Synthetic** *Human serum albumin (HSA) and polyethylene glycol (PEG)*	The combination of PEG and HSA allows for the hydrogel to effectively seal or reduce air leaks in the lungs. Its elasticity enables it to move with the lungs during breathing, while its adhesiveness ensures a secure seal.	Pleural air leak sealant.	Approved—FDA 2010
22	**Natural** *Collagen*	A denatured porcine collagen implant works by stimulating the body’s own collagen production and promoting an inflammatory response to fill in depressed areas like scars and wrinkles. It achieves this through soft tissue augmentation, effectively lifting the depressed area.	Treatment of depressed areas like scars and wrinkles.	Approved—FDA 1988
23	**Natural** *Collagen, carboxymethylcellulose, and recombinant* OP-1	Collagen provides a scaffold, CMC enhances handling and may improve bone formation, and rhOP-1 is a protein that stimulates bone growth. This combination is used in devices, like “OP-1 putty”, for bone grafting and spinal fusion.	Bone regeneration and spinal fusion.	Approved—FDA 2001
24	**Natural** *Bovine collagen*	Collagen provides a scaffold for cell adhesion and tissue ingrowth, ultimately leading to correction of soft tissue defects. The crosslinking of the collagen with glutaraldehyde makes it more resistant to immediate degradation than non-crosslinked collagen, allowing for a longer-lasting correction.	Facial volume restoration; correction of contour deficiencies.	Approved—FDA 1981
25	**Natural** *Human collagen*	It works by directly replenishing lost collagen in the skin, which helps to fill in wrinkles and other depressions caused by aging or other factors. It does this by integrating into the skin’s structure and providing structural support, similar to the body’s own collagen.	For wrinkles and acne scars correction; correction of soft tissue contour deficiencies.	Approved—FDA 2003
26	**Natural** *Hyaluronic acid with lidocaine*	The crosslinked hyaluronic acid fills moderate/severe facial wrinkles and folds. The Cohesive Polydensified Matrix (CPM) technology allows for even distribution and integration into the skin, resulting in a natural-looking and smooth correction.	(i) Facial wrinkles and folds corrections. (ii) For the treatment of tear troughs.	Approved—FDA 2006, 2012, 2018 and 2019 Approved—FDA 2023
27	**Natural** *Hyaluronic acid*	It works by replenishing lost volume and smoothing wrinkles through its HA component. The HA binds with water molecules, hydrating the skin and creating a plumping effect. (ii) Restoration of face volume by forming a structural support due to crosslinking of HA. (iii) The HA administered along the jawline and chin provides immediate structural support, lifting the jawline and improving definition. This also helps smooth out wrinkles and fine lines, creating a more sculpted look.	(i) Facial wrinkles and folds corrections. (ii) Lip filler; facial volume corrections. (iii) Jawline definition.	Approved—FDA 2017 Approved—FDA 2020 Approved—FDA 2022
28	**Natural** *Hyaluronic acid and* *1,4-butanediol diglycidyl* *ether with lidocaine*	1,4-butanediol diglycidyl ether is a crosslinking agent used to bind HA molecules together, creating a highly stable and durable gel. HA crosslinked with 1,4-butanediol diglycidyl ether (BDDE) provides smoothness and hydration without adding significant volume. The 0.3% lidocaine component reduces pain during the injection.	(i) Improves the smoothness and hydration of the skin. (ii) For the correction of moderate-to-severe dynamic perioral rhytids.	Approved—FDA 2023 Approved—FDA 2017 and 2021
29	**Natural** *Hyaluronic acid with calcium phosphate (CaP)*	HA’s ability to attract water creates a favorable microenvironment for cell attachment and proliferation, and CaP’s role promotes bone formation and mineralization.	Bone void filler for orthopedic applications.	Approved—FDA 2019
30	**Natural** *Cinnamic acid-functionalized hyaluronic acid*	It acts as a viscoelastic supplement to the synovial fluid, aiming to restore the natural lubricating and shock-absorbing properties of a joint, reduce pain and inflammation, and potentially offer some level of cartilage protection.	For treatment of osteoarthritis.	Approved—FDA 2011

## Data Availability

The data presented in this study are available upon request from the corresponding author. The data are not publicly available due to ethical considerations.

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
