# Peer review of "Functional Hydrogels: A Promising Platform for Biomedical and Environmental Applications"

_ijms, 2025, doi:10.3390/ijms26189066_

Round 1

Reviewer 1 Report

Comments and Suggestions for Authors

The manuscript on “Functional Hydrogels: A Promising Platform for Biomedical and Environmental Applications” presents a relevant topic with potential value for the field. The review is well structured and outlined the Rational design strategies, Biomedical Applications, Environmental and Soft Robotics Applications along with Challenges, Limitations, and Safety Considerations and Future Perspectives. However, certain sections, especially introduction, need more clarity and alignment with the main subject functional hydrogels. The authors can also add a subpoint separately such as synthesis techniques of functional hydrogels.

  • The introduction part of the manuscript spends too much information on nanotechnology and nanoparticles drawbacks before introducing functional hydrogels properly.
  • Authors are suggested to add why hydrogels are advantageous over other material platforms under introduction section.
  • Include recent biomedical and environmental applications to demonstrate the link under introduction section.
  • The nanohydrogel discussion is totally biomedical focused under Rational design strategies for functional hydrogels. Environmental applications doesn’t connect with this concept. Authors are suggested to explore nanohydrogels strategies for environmental applications as well.
  • The biomedical and environmental sections are presented without any link. Authors are suggested adding a bridging section to make the hydrogels equally suitable for both.
  • Although heavy metal removal is discussed in detail under Environmental and Soft Robotics Applications of functional hydrogel, other environmental applications mentioned in the abstract such as pollutant detection and environmental sensing are not covered broadly.

Reviewer 2 Report

Comments and Suggestions for Authors

This review comprehensively addresses functional hydrogels’ rational design (natural/synthetic/hybrid polymers, physical/chemical cross-linking), multi-field applications (biomedical, environmental heavy metal removal, soft robotics), clinical translation (FDA-approved cases), challenges and future directions. With a clear structure and rich citations, it offers a holistic field overview. However, it needs optimization in application coverage, content depth and detail presentation before publication.

  1. Insufficient Depth in Clinical Translation Analysis: The review lists FDA-approved clinical cases in Table 1 but lacks comparative analysis of core differences (translation success rate, long-term safety, adverse reactions, production cost) between natural (e.g., hyaluronic acid, collagen) and synthetic (e.g., PEG, PLLA) polymer hydrogels. It is recommended to supplement this dimensional analysis using Table 1 data to clarify their pros, cons, and suitable clinical scenarios.
  2. The review’s environmental section only covers heavy metal ion (e.g., Hg²⁺, Pb²⁺, U⁶⁺) removal/detection, omitting functional hydrogel progress in treating other pollutants (organic dyes, antibiotics, microplastics)—which are major environmental hazards with related hydrogel studies available. Recommend supplementing this content and elaborating on pollutant-specific design strategies (e.g., surface modification, responsive adsorption).
  3. The review has key figures (e.g., Fig 1, 3, 5) but no concise textual explanation of their core info (e.g., cross-linking principle, Hg²⁺ detection signal logic) in the main text. Newcomers may struggle to understand. Recommend adding 1-2 sentences per figure to explain key details (e.g., Fig 5’s SYBR Green I fluorescence-Hg²⁺ correlation, Fig 3’s cross-linker effect on HA hydrogel stability) for better figure-text connection.
  4. The "Challenges and Limitations" chapter of the review mentions common issues such as biocompatibility and mechanical stability, but fails to analyze scenario-specific challenges in combination with specific application scenarios (e.g., in vivo drug delivery, in vitro tissue engineering scaffolds, environmental water purification).
  5. The "Future Perspectives"mentions bioorthogonal chemistry, 3D/4D printing, AI-assisted design but no recently publication specific cases (e.g., AI hydrogel design algorithms, 3D-printed tissue scaffolds). Recommend adding representative studies (e.g., AI-optimized hydrogel degradation, 4D-printed hydrogels, etc.) to clarify progress and breakthrough points.

Reviewer 3 Report

Comments and Suggestions for Authors

General remark. 
The article, as it is written now, boils down to a list of individual published works in the field of hydrogel materials. The work does not sufficiently analyze the development of these works in certain areas, purposefully does not highlight the possible solution to research problems. The way of writing of the material is not structured enough, which does not allow you to navigate it. The article very often uses the phrase about the great potential of hydrogels in biomedical applications, but this potential is definitely not a general phrase, not formulated and the development of this potential is not analyzed.

It is necessary to clarify what the authors mean by the term "functional hydrogels". It is whether a hydrogel gel-forming polymer that contains functional groups or a hydrogel that generally plays a role in a specific use.
It is advisable for authors to structure the presentation of the material more clearly. In lines 31 ÷ 65, the authors give a brief overview of the problems of nanodisperse therapeutic delivery systems with a fair emphasis on their problems. Lines 87÷108 emphasize that these problems can be largely overcome by the use of nanogels. It is assumed that the properties of nanogels and methods of their production will be analyzed in the future, but in the article the authors move on to the general analysis of gels and the advantages of nanogels remain uncertain.
The biomedical application of hydrogels in Chapter 2 is described too conceptually. It boils down to enumerating directions without analysis. The reasons for which the analysis of works on hyaluronic acid is separately selected remain unclear. This section is appropriate to specify so that its generality can be reduced for a more detailed description of the problems of using hydrogels.  All these areas ( listed in line 162, tissue creation, lubricants, wound dressings, contact lenses, and drug delivery ) differ significantly in terms of the hydrogel materials used in them. The given individual solutions in the section do not allow us to conclude about problems in application in any of these areas.
The informational value of Figure 7 is highly controversial. By capture, these are types of hydrogels. Hydroxyapatite is an inorganic salt that can be used in bone regeneration preparations, but is not a hydrogel. What the authors mean by differentiation, within this scheme, human collagen, porcine collagen and simply collagen.
The expediency within the scope of this article of Chapter 3 is doubtful. It is doubtful to combine in one small section at the same time the problems of using hydrogels in the absorption of heavy metals, the design of sensors based on hydrogel and robotic systems. None of these problems are at the development level, restrictions on use are not open, development prospects are not indicated. The presented material boils down to a list of individual, new, solutions, but not those that are a real development of problems. 
Section 4 is important and valuable. Thanks to the materials of Table 1, it is possible to form an impression of the progress of various types of hydrogels in clinical practice. In-depth analysis of the data of this table allows us to understand the stages of development of hydrogel materials in clinical practice. 
Chapter 5 fully describes the problems of hydrogel development in the biomedical field. An extensive discussion of the issues raised in this section is the subject of a review article on the use of hydrogels in medicine. But, according to the text of the article, the authors limited themselves only to declaring these problems, which are already generally known. A systematic analysis of ways to solve these problems based on the analysis of individual publications is of interest for the review article.

Round 2

Reviewer 2 Report

Comments and Suggestions for Authors

Regarding the reviewer's comment mentioned previously: "The review’s environmental section only covers heavy metal ion (e.g., Hg²⁺, Pb²⁺, U⁶⁺) removal/detection, omitting functional hydrogel progress in treating other pollutants (organic dyes, antibiotics, microplastics)—which are major environmental hazards with related hydrogel studies available. Recommend supplementing this content and elaborating on pollutant-specific design strategies (e.g., surface modification, responsive adsorption)", no relevant discussions on the progress of functional hydrogels in treating other pollutants (organic dyes, antibiotics, microplastics) have been found in the revised manuscript.

Author Response

Our response: We thank the reviewer for the helpful comments. We have revised the section, and we believe this change has improved our manuscript well. (pp. 15 & pp. 16 lines 321-365)

Reviewer 3 Report

Comments and Suggestions for Authors

Thank you for your responses to my comments. They allowed me to better
understand your intentions when writing the manuscript. After you
reviewed the manuscript, its text and the conclusions that you intended
to convey to the reader became clearer. Thanks to the revisions, reading
and assimilation of information became easier. However, I cannot say
that the article has become significantly more informative for
professional scientists whose research interests lie in the field of
creating new hydrogel materials. As in the first version, I believe that
it lacks sufficient analysis and depth of review for a review article.
However, the manuscript provides a broad overview of the areas of
application of hydrogels, describes interesting solutions and reviews
new publications. Therefore, this manuscript is of some practical
interest to scientists who assess the feasibility of certain projects.
Therefore, I will recommend your work for publication with minor
corrections. Regarding minor corrections:
1. It is necessary to clarify in the abstract what the authors consider
to be “functional” hydrogels. In particular, which hydrogels do the
authors consider to be “non-functional”?
2. It is also necessary to clarify the facts stated. For example, in
lines 100-102, by which parameter do microhydrogels prevail over
liposomal and polymeric particles according to the data of 72?
3. In section 1, especially when analyzing the diagrams in Fig. 2, it is
advisable to indicate which methods are “rational” for the synthesis of
microhydrogels, as indicated in the title of the section.
4. In line 140, it is advisable to clarify what the authors mean by the
term “neutral” charges. The use of this term can be considered a
typographical error, but it is advisable to indicate why neutral surface
charges are effective and in which works this is shown.

Author Response

Response to Reviewer 2

Comments 1: It is necessary to clarify in the abstract what the authors consider to be “functional” hydrogels. In particular, which hydrogels do the authors consider to be “non-functional”?.

Our response: We thank the reviewer for this important suggestion. In the revised abstract, we now provide a concise definition of functional hydrogels as “Functional hydrogels are deliberately engineered with specific chemical groups, stimuli-responsive motifs, or crosslinking strategies that impart targeted biomedical or environmental roles (e.g., drug delivery, pollutant removal). In contrast, conventional or “non-functional” hydrogels are defined as passive polymer networks that primarily serve as water-swollen matrices without such application-oriented modifications.” This clarification ensures that readers can immediately understand our intended usage of the term. Moreover, we do not propose the term non-functional, but the earlier researchers suggest that the functional hydrogels are “functional based on their multifunctionalities. (Section Abstract, pp. 1, lines 10-16)

            “Hydrogels have great potential to present multi-functional properties. Different functional aspects, such as biocompatibility, biodegradability, adhesiveness, vascularization potential, as well as antimicrobial, anti-inflammatory, and pro-angiogenic properties, can be incorporated into hydrogels for chronic wound healing.”

Similarly, several other reports suggest the use of the term "functional hydrogel" for the hydrogels intended to be used in multidimensional roles with intrinsic properties/functionalities or by adhering multiple functional moieties or functional groups.

Reference –

1) Firlar, I.; Altunbek, M.; McCarthy, C.; Ramalingam, M.; Camci-Unal, G. Functional Hydrogels for Treatment of Chronic Wounds. Gels 20228, 127. https://doi.org/10.3390/gels8020127

2) Yifan Liu.; Zhiguang Guo. Small functional hydrogels with big engineering applications. Materials Today Physics, Volume 43, 2024, 101397, https://doi.org/10.1016/j.mtphys.2024.101397

Comments 2: . It is also necessary to clarify the facts stated. For example, in lines 100-102, by which parameter do microhydrogels prevail over liposomal and polymeric particles according to the data of 72?

Our response: We thank the reviewer for the helpful comment. We appreciate the reviewer’s request for clarification. We apologize for the inconvenience; the relevant citations were misplaced. Following the reviewer’s helpful comment, we have revised the section with the relevant citations. (pp.3 lines 104-105)

“Hydrogels hold significant potential in regenerative medicine and drug delivery, offering advantages such as excellent biocompatibility, water retention, and the ability to mimic natural tissues, which can surpass the capabilities of standard nanocarriers like liposomes and micelles alone. They serve as scaffolds for tissue engineering, control the release of drugs, and can even protect encapsulated drugs within their porous structure. Hydrogels demonstrated superiority in terms of drug loading capacity and controlled release kinetics compared with liposomal and conventional polymeric nanoparticles. This distinction highlights the enhanced efficiency of hydrogels as carriers for drug delivery.” Moreover, we would like to draw the reviewers' attention to reference 78 to support our statement in the manuscript.

References –

1) Advances in drug delivery systems based on liposome-composite hydrogel microspheres, J. Mater. Chem. B, 2025, https://doi.org/10.1039/D5TB01369K

2) D. E. Large, R. G. Abdelmessih and E. A. Fink, et al., Liposome composition in drug delivery design, synthesis, characterization, and clinical application, Adv. Drug Delivery Rev., 2021, 176, 113851, DOI:10.1016/j.addr.2021.113851

Comments 3: In section 1, especially when analyzing the diagrams in Fig. 2, it is advisable to indicate which methods are “rational” for the synthesis of microhydrogels, as indicated in the title of the section.

Our response: We thank the reviewer for this constructive observation. We have revised section 1 and the discussion around Fig. 2 to explicitly mention the rational design strategies (pp. 3 & pp. 4, lines 116-125). Following the reviewer's comment, we have revised the section. We believe these clarifications make the figure-text connection clearer, and this change has improved our manuscript well.

Comments 4: In line 140, it is advisable to clarify what the authors mean by the term “neutral” charges. The use of this term can be considered a typographical error, but it is advisable to indicate why neutral surface charges are effective and in which works this is shown.

Our response: We are grateful to the reviewer for noting this ambiguity. We have clarified that by “neutral charges,” we refer to “zwitterionic or charge-balanced surface modifications that result in an overall neutral surface potential”. Such surfaces are effective because they minimize nonspecific protein adsorption and reduce immune recognition, thereby improving circulation time in vivo.

            Following the reviewer’s helpful comment, we have added a short description in brackets to explain to the readers what the author means by the term “neutral charges”. We have cited the studies in the relevant section, and along with this, we have provided the link below. We truly thank the reviewer for highlighting this discrepancy. We believe that after following the revision, our manuscript has improved well.

(“zwitterionic or charge-balanced surface modifications that result in an overall neutral surface potential.”)

References –

1) Engineering principles of zwitterionic hydrogels: Molecular architecture to manufacturing innovations for advanced healthcare materials, Materials Today Bio, Volume 33, August 2025, 102085 https://doi.org/10.1016/j.mtbio.2025.102085

2) Zwitterionic hydrogels from material design to wound dressing applications, Supramolecular Materials, Volume 4, December 2025, 100108, https://doi.org/10.1016/j.supmat.2025.100108